# Microstructure Evolution and Mechanical Properties of FeCoCrNiCuTi_0.8_ High-Entropy Alloy Prepared by Directional Solidification

**DOI:** 10.3390/e22070786

**Published:** 2020-07-18

**Authors:** Yiku Xu, Congling Li, Zhaohao Huang, Yongnan Chen, Lixia Zhu

**Affiliations:** 1School of Material Science and Engineering, Chang’an University, Xi’an 710064, China; 2019131046@chd.edu.cn (C.L.); 2017231005@chd.edu.cn (Z.H.); chenyongnan@chd.edu.cn (Y.C.); 2CNPC Tubular Goods Research Institute, Xi’an 710077, China; zhulx@cnpc.com.cn

**Keywords:** high-entropy alloy, directional solidification, crystal structure, primary dendrite arm spacing, mechanical property

## Abstract

A CoCrCuFeNiTi_0.8_ high-entropy alloy was prepared using directional solidification techniques at different withdrawal rates (50 μm/s, 100 μm/s, 500 μm/s). The results showed that the microstructure was dendritic at all withdrawal rates. As the withdrawal rate increased, the dendrite orientation become uniform. Additionally, the accumulation of Cr and Ti elements at the solid/liquid interface caused the formation of dendrites. Through the measurement of the primary dendrite spacing (λ_1_) and the secondary dendrite spacing (λ_2_), it was concluded that the dendrite structure was obviously refined with the increase in the withdrawal rate to 500 μm/s. The maximum compressive strength reached 1449.8 MPa, and the maximum hardness was 520 HV. Moreover, the plastic strain of the alloy without directional solidification was 2.11%, while the plastic strain of directional solidification was 12.57% at 500 μm/s. It has been proved that directional solidification technology can effectively improve the mechanical properties of the CoCrCuFeNiTi_0.8_ high-entropy alloy.

## 1. Introduction

Conventional alloys are designed by using one or two principle elements and adding other trace elements. In 2004, it was reported that the FeCoCrNiMn alloy has only a single phase as a multicomponent alloy [1]. Yeh et al. calls such alloys high-entropy alloys (HEAs) and proposes new alloy design concepts [2]. These alloys are composed of five or more elements in equal or nearly equal molar proportions. Multi-principle high-entropy alloys have four special effects [3]: high-entropy effect, lattice distortion effect, sluggish diffusion effect, and cocktail effect. The high-entropy effect makes the alloy ultimately crystallize into a solid solution rather than intermetallics, and the lattice distortion effect further improves the material properties. In the past decade, a variety of HEAs have been reported, and many excellent properties have been found, such as excellent compression or stretch performance [4,5], good wear performance [6,7], ferromagnetic properties [8], creep resistance [9], biocompatibility [10], deformation behavior [11], and excellent mechanical and magnetic properties [12,13]. 

In the initial research, the HEAs were considered to be a single-phase solid solution. However, this is considered very difficult [14]. At the same time, single-phase solid solutions often have good performance in only one aspect, and it is difficult to achieve a balance between plasticity and strength. Therefore, in recent years, many HEAs based two-phase or multi-phase alloys, such as CoCrFeNiAlx [15], CoCrFeNiCuAlx [2], AlCoCrFeNi_2.1_ [16], etc., have been reported. By adjusting the hexagonal close-packed (HCP)/topological close-packed (TCP)/body-centered cubic (BCC) phases in these multiphase HEAs, the performance of HEAs can be effectively improved.

The CoCrCuFeNi high-entropy alloy has a single-phase face-centered cubic (FCC) structure with high toughness [17]. Forming the second phase by adding Ti or Al can improve the mechanical properties of CoCrCuFeNi. AlxCoCrCuFeNi (x = 0.5, 1.5, 2.5, and 4) high entropy alloys were synthesized by mechanical alloying and spark plasma sintering. Due to the addition of aluminum, the microstructure of the material evolves from FCC and BCC phases to FCC, BCC, and ordered BCC phases. In addition, with the increase in Al content, the further synergistic effect of strength and ductility and more obvious strain hardening are observed [18]. Aluminum also plays an important role in high temperature resistance. The high temperature oxidation resistance of the AlxCoCrCuFeNi (x = 0, 0.5, 1, 1.5, 2) high-entropy alloy was studied. The results show that the oxidation resistance of the Al_2_CoCrCuFeNi alloy is the best among the five HEAs [19]. A previous report has indicated that the crystal structure of CoCrCuFeNiTix is: FCC (x = 0, 0.5) and FCC + Laves phases (x = 0.8, 1). With the addition of Ti, the strength of the CoCrCuFeNiTix alloy increases but the ductility decreases [20]. In order to obtain a good combination of strength and ductility, both FCC and Laves phases were required. Therefore, in this study, a CoCrCuFeNiTi_0.8_ alloy with a dual-phase structure was selected.

It is worth noting that the properties of high-entropy alloys depend not only on their composition, but also on the processing technology. A variety of technologies have been used to prepare HEAs, such as thermal spray (TS) [21], laser cladding [22], and mechanical alloying (MA) [23]. Directional solidification (DS) technology has been used to prepare aero-engine turbine blades [24], and the mechanical properties of materials have been improved [24,25,26]. For example, the Ni-based superalloy IN738LC obtained a well-oriented dendritic structure after directional solidification, which improved its tensile properties [27]. At present, DS technology has been used to prepare HEAs, single crystal HEAs have been successfully prepared, and the performance of HEAs has been improved [28,29]. However, there are few studies on two-phase high-entropy alloys with FCC + Laves.

In this study, a high-entropy alloy of FeCoCrNiCuTi_0.8_ was prepared by DS, and the microstructural transformation and mechanical properties were studied by different microscopic observation techniques (optical microscope, scanning electron microscope, and transmission electron microscope). The analysis of segregation of different elements in high-entropy alloy was performed using energy dispersive spectrometry. In addition, the compressive properties of the high-entropy alloy were analyzed according to the microstructure at different withdrawal rates.

## 2. Materials and Methods

In this work, the CoCrCuFeNiTi_0.8_ alloy (nominal composition) was investigated. Granular Co, Cr, Cu, Fe, Ni, and Ti (purity > 99.9%) were selected as raw materials. It is worth noticing that the raw materials needed to be washed with alcohol and acetone to remove surface impurities and grease before use. Button ingots were prepared through an electric arc furnace (KDX-600). The melting process needed to be performed in an argon atmosphere to avoid the oxidation of the raw materials. They also needed to be remelted five times to ensure uniform composition. The button ingot was remelted using a vacuum induction furnace (ZG-0.025) to obtain a master alloy of Φ30*80 mm. The master alloy was cut into Φ7*80 mm rod-shaped samples by a wire electro-discharged machine. Under a 200 K/cm temperature gradient, DS was completed using a Bridgeman furnace at different withdrawal rates (50, 100, and 500 μm/s).

Metallographic samples were prepared by first grinding with 70 to 2000 mesh SiC abrasive paper, and then polishing with Cr_2_O_3_ powder. The metallographic samples needed to be corroded with aqua regia for 20 seconds before observation. The microstructure of the sample was investigated through an Axio Scope-A1 optical microscope (OM) with a Carl Zeiss digital camera and an S-4800 scanning electron microscope (SEM). The primary dendrite arm spacing was measured using the minimum spanning tree method. The phase composition was analyzed by X-ray diffraction (XRD) using a Bruker D8 Advance model, using Cu–Kα (λ = 0.15406nm) radiation scanning from 20° to 90° at a scanning rate of 8°/min. Using a standard sized smooth tube, the working voltage was 40kV and the current was 40mA. In order to prepare transmission electron microscope (TEM) samples, a 1-mm-thick thin section was cut from the cross section of the sample, and then the sample was ground to 80 μm with abrasive paper. Finally, the thin area below 100nm was prepared by an ion thinning instrument for observation. The microstructure was observed using TEM and its elemental composition was analyzed using energy dispersive spectrometry (TEM-EDS, FEI Talos F200X). Microhardness was measured using a Vickers hardness tester (HX-1000TM) at a load of 100 g. The values stated in this study are the averages of at least five different indentations scattered over the sample. The distance between the centers of two adjacent indentations was at least three times the diagonal length of the indentation. The distance from the center of any indentation to the edge of the sample was not less than 2.5 times that of the diagonal length of the indentation. Compressive properties were studied using a universal mechanical testing machine (MTS) at a strain rate of 10^−4^/s.

## 3. Results and Discussion

### 3.1. Microstructure Evolution

Figure 1 shows the microstructure of DS CoCrCuFeNiTi_0.8_ at the withdrawal rates of 50 μm/s, 100 μm/s, and 500 μm/s. The longitudinal microstructure of the samples at different withdrawal rates is shown in Figure 1b–d. It can be seen that there are three stages of directional solidification growth: initial stage, transition stage, and stable stage. In the transition stage, some large, black, round particles can be observed, which may be caused by the excessive corrosion of Cu. In the stable stage, these round particles become finer, and the Cu-rich particles formed by Cu segregation are clearly distributed on the solidified structure. Compared with the transition stage, the dendrite structure with a uniform orientation can be observed in the stable stage, which shows that directional solidification promotes the preferred orientation of grains. That is to say, with the progress of solidification, the orientation of dendrites tends to be consistent. The orientation of dendrites quickly becomes uniform at the stable stage of 100 and 500 μm/s compared to 50 μm/s, in other words, the large withdrawal rate makes the transition stage narrower.

It can be seen from Figure 1e–g that the DS structure of the alloy is dendritic and interdendritic. Comparing the DS structure at different withdrawal rates, the 50 µm/s sample is obviously loosely distributed. When the withdrawal rate increases to 500 μm/s, the dendrite structure becomes finer. Therefore, the larger withdrawal rate makes the microstructure finer.

The microstructures of the stable regions of the directional solidification samples at different withdrawal rates are exhibited in Figure 2. It can be seen from the figure on the left side that the consistency of dendrite orientation is poor at 50 μm/s, while the sample with a withdrawal rate of 500 μm/s solidifies in the stable region with almost the same dendrite orientation. Therefore, higher withdrawal rates can result in a consistent dendrite orientation. The enlarged figure on the right indicates that the primary dendrite spacing of the alloy is larger at 50 μm/s, and the dendrite structure is finer at 500 μm/s. Combined with Figure 1, it can be deduced that the larger the withdrawal rate, the finer the dendrite structure.

It can be seen from Figure 3 that the primary phase is a cross-shaped dendrite, and the secondary dendrite arms are developed. The interdendritic structure is a clear two-phase eutectic structure. The dendrite morphology is shown in Figure 3d. The most important parameter for the formation of microstructure during directional solidification is the scale feature. For the dendrite array formed by directional solidification, this scale is the primary dendrite spacing. For the directional solidification structure with developed secondary dendrites, the primary dendrite spacing (λ_1_) and the secondary dendrite spacing (λ_2_) are very important parameters. They correspond to λ_1_ and λ_2_ in Figure 3a,c, respectively.

### 3.2. Structural Parameters 

The values of the primary dendrite arm spacing (λ_1_) and secondary dendrite arm spacing (λ_2_) are measured on the transverse sections, respectively. The MST method has been proven to accurately measure the dendrite spacing [30]. MST refers to an unclosed figure formed by connecting all points, and the total path of the resulting spanning tree must be minimized, so the generated figure can represent the minimum total length of connecting all points. Therefore, the MST method is used to measure λ_1_.

The dendrite centers in Figure 4a are connected to obtain Figure 4b, and the distance between the adjacent points is calculated. Obviously, this statistical result fits the Gaussian distribution. The fitting coefficient R^2^ of the statistical result, calculated according to Figure 4c, is 0.98, which is close to 1. It shows that the statistical results are more accurate. The secondary dendrite arm spacing is directly measured from the cross-sectional micrograph. Figure 4d shows λ_1_ and λ_2_ at different withdrawal rates. It can be seen that, as the withdrawal rate increases, λ_1_ and λ_2_ gradually decrease. It is further confirmed that the microstructure is refined with the increase in withdrawal rate.

### 3.3. Phase Composition and Segregation Analysis

Figure 5 shows the XRD spectra of DS CoCrCuFeNiTi_0.8_ at different withdrawal rates. The XRD spectrum confirms the existence of an FCC phase and a Laves phase in DS CoCrCuFeNiTi_0.8_. The lattice parameter of the FCC phase is 3.579 Å. The lattice parameters of the Laves phase are a = 4.765 Å and c = 7.723 Å, and it should be Fe_2_Ti type. In addition, it can be seen from the figure that, in the process of directional solidification, the constituent phase of the structure has no change at different withdrawal rates.

Figure 6a exhibits an obviously eutectic structure with two phases. Combining Figure 6b and its EDS map, we can see the element distribution of two phases. The diffraction spots in Figure 6j correspond to the C15 structure of an Fe_2_Ti-type Laves phase with a zone axis of [001]. The diffraction spots in Figure 6l correspond to the FCC phase with a zone axis of [212]. The crystal surface spacings are 0.331 nm and 0.127 nm in Figure 6i,k, respectively. From the element distribution of the EDS mapping, Cr, Fe, and Ni are mainly distributed on the dendrite. Co and Ti elements are mainly distributed on the interdendritic structure. Previous studies have shown that the the FCC phase is directly precipitated from the liquid phase [19]. Laves is produced by a eutectic reaction in the liquid phase. Therefore, it can be considered that the primary dendrite is an FCC phase, and the interdendritic structure is a eutectic structure composed of FCC + Laves. At the same time, a large amount of Cu is segregated at the two-phase interface because Cu is not stable in FCC dendrites and is excluded from the dendrite region.

The experimental results show that the microstructure of the directional solidification sample consists of dendrites (FCC) and interdendritic structures (FCC + Laves). During the solidification process, the FCC phase was first precipitated from the liquid phase at 1175 °C. Subsequently, the eutectic reaction of the remaining liquid phase occurred at 1050 °C to generate the eutectic structure of the FCC + Laves phases. It has been reported that the diffusion rate of Cu is faster than that of other elements [31], and the melting point of Cu is also lower, which may promote the segregation of Cu at the phase boundary, and the content of Cu in the two phases is relatively low. The interdendritic composition was calculated as the average composition of two phases. The composition of the microstructure is shown in Table 1.

The segregation of elements based on the segregation ratio $k^{\prime }$ was calculated, which is specifically defined as follows:(1)$$k^{\prime }=C_{DC}/C_{ID},$$
where $C_{DC}$ is the element concentration in the lamellar dendrite and $C_{ID}$ is the element concentration in the interdendritic region. When $k^{\prime }$ is less than 1, this element is segregated. The calculated $k_{Cr}^{\prime }$ and $k_{Ti}^{\prime }$ are less than 1, indicating that Cr and Ti have severe segregation. During directional solidification, Cr and Ti accumulate in front of the solid/liquid interface. Because of the slow diffusion effect of high-entropy alloys, the segregation of these elements is difficult to alleviate by diffusion. Additionally, the segregation of elements results in large constitutional supercooling. The interface instability leads to the formation of the dendrite structure.

### 3.4. Mechanical Behavior 

The microhardness and primary dendrite arm spacing at different withdrawal rates are demonstrated in Figure 7a. As can be seen, with the increase in the withdrawal rate, the microhardness increased from 415.5 HV to 466 HV and finally to 520 HV. The room temperature compression curve is shown in Figure 7b. No apparent yielding platform is seen in the curve. The samples show good compressive plasticity, and the compressive strain is greater than 10% at different withdrawal rates. At the same time, as the withdrawal rate increases, the compressive strength increases from 1237.3 MPa (V = 50 μm/s) to 1421.2 MPa (V = 100 μm/s), and then to 1449.8 MPa (V = 500 μm/s). Previous studies have proven that improving the performance of HEAs by changing the synthesis process is a very effective means. The microstructure of the CoCrCuFeNiTi_0.8_ HEA can be controlled by changing the withdrawal rate. In this experiment, the alloy achieved the maximum compressive strength of 1449.8 MPa at 500 μm/s. This is mainly due to the refinement of the dendrite structure. At high withdrawal rates, the diffusion of Cr and Ti elements is suppressed, and a larger structural supercooling is obtained. This causes more nucleation during the solidification process, resulting in a finer dendritic structure. During the deformation process, moving dislocations tend to accumulate at the dendrite–interdendritic interface. Therefore, the interface between the dendrites and the interdendritic structure is considered an obstacle to dislocation movement. In the refined dendrite structure, there are more interfaces in the same volume. Therefore, the alloy obtained the maximum compressive strength in a room temperature compression test at 500 μm/s.

Figure 7c shows the compressive properties of the CoCrCuFeNiTi_0.8_ alloy in different states. It has been previously reported that the compressive strength of conventional as-cast CoCrCuFeNiTi_0.8_ alloys is 1848 MPa and the plastic strain is 2.11% [20]. It can be seen that the compressive strength of the alloy decreases slightly after directional solidification under different withdrawal rates. The plastic strain increased to about 11%. It shows that directional solidification can effectively improve the plasticity of the CoCrCuFeNiTi_0.8_ alloy. This similar result is also reported for the DS AlCoCrFeNi_2.1_ eutectic high-entropy alloy [32]. The good combination of strength and plasticity is obtained, and the machinability of the alloy is improved. The refinement of the CoCrCuFeNiTi_0.8_ alloy’s microstructure leads to an increase in plasticity. During plastic deformation, there are more grains that are favorable for sliding, and the deformation is evenly dispersed across more grains. In addition, the finer the structure, the more tortuous the interface and the weaker the microcrack propagation. These are conducive to improving the plastic deformation ability of the metal. 

In directional solidification studies of other alloys, it has also been reported that the refined dendritic structure has better compression properties. This includes Al-Ni [33], Cu-Ce [34], and CoCrFeNiAl_0.7_ [29]. Reducing the primary dendrite spacing and optimizing the shape and distribution of the second phase can make the alloy obtain better mechanical properties. Other compression performance data are shown in Table 2.

Figure 8 demonstrates the fracture morphology of the directionally solidified CoCrCuFeNiTi_0.8_ high-entropy alloy. An obvious cleavage fracture mode is observed in all alloys, and the cleavage surface is of the river pattern. The fracture morphology of the sample prepared at 50 μm/s is shown in Figure 8a. Some cleavage steps are observed. From the magnified image (Figure 8a_1_), it can be observed that the cleavage steps are arranged in order with the same direction, which indicates the difficulty of crack growth. Figure 8b shows the fracture morphology of the alloy with small cracks at 100 μm/s, and the brittle fracture of the material is caused by the high level of segregation of Cu at the phase boundary. A brittle massive stacking fault is observed, as displayed in Figure 8b_1_, which determines the cleavage fracture mode. When the withdrawal rate is further increased to 500 μm/s, as shown in Figure 8c, the fracture surface is relatively flat and small cracks are also observed. Additionally, the tongue pattern is observed in the magnified image (Figure 8c_1_). To sum up, the fracture mode of the alloy has no obvious change at different withdrawal rates, all of them are brittle fractures. With the increase in withdrawal rate, the microstructure is refined and the crack growth resistance of the alloy is improved. This results in a higher strength of the alloy.

## 4. Conclusions

The CoCrCuFeNiTi_0.8_ HEA was prepared by directional solidification at different withdrawal rates (50 μm/s, 100 μm/s, 500 μm/s). The microstructure, crystal structure, and mechanical properties were investigated. The following conclusions can be drawn from this study.

(a) The microstructure of the directional solidified alloy is a dendritic structure. Within it, the dendrite is composed of an FCC phase. The interdendrite is a eutectic structure composed of FCC phase and Laves.

(b) During solidification, Co and Ti elements are enriched in the interdendritic region. Cr, Fe, and Ni are enriched in dendrites. The enrichment of Cr and Ti elements at the front of the solid/liquid interface promotes the formation of dendritic structures. At the same time, a high level of segregation of Cu was found at the two-phase interface.

(c) With the increase in withdrawal rate from 50 μm/s to 500 μm/s, the primary dendrite arm spacing decreased from 187.6 μm to 118.2 μm. The secondary dendrite arm spacing decreased from 30.7 μm to 16.5 μm. The refinement of the dendrite structure effectively improved the compressive properties of the alloy. The maximum compressive strength was 1449.2 MPa at 500 μm/s.

(d) The plastic strain of the alloy increased from 2.11% in the conventional as-cast state to 12.75% after directional solidification at 500 μm/s. Directional solidification can effectively improve the compressive properties of the alloy. The fracture surfaces of the directionally solidified CoCrCuFeNiTi_0.8_ high-entropy alloy showed a cleavage fracture mode.

In conclusion, the microstructure of directionally solidified alloy is a dendrite structure and the enrichment of Cr and Ti elements at the front of the solid/liquid interface promotes the formation of dendritic structures. With the increase in withdrawal rate, the dendrite orientation tends to be uniform and the dendrite structure is obviously refined. The plasticity of the directionally solidified CoCrCuFeNiTi_0.8_ high-entropy alloy is much higher than that of a conventional as-cast alloy. It has been proved that directional solidification technology can effectively improve the mechanical properties of the CoCrCuFeNiTi_0.8_ high-entropy alloy.

## Figures and Tables

**Figure 1 entropy-22-00786-f001:**
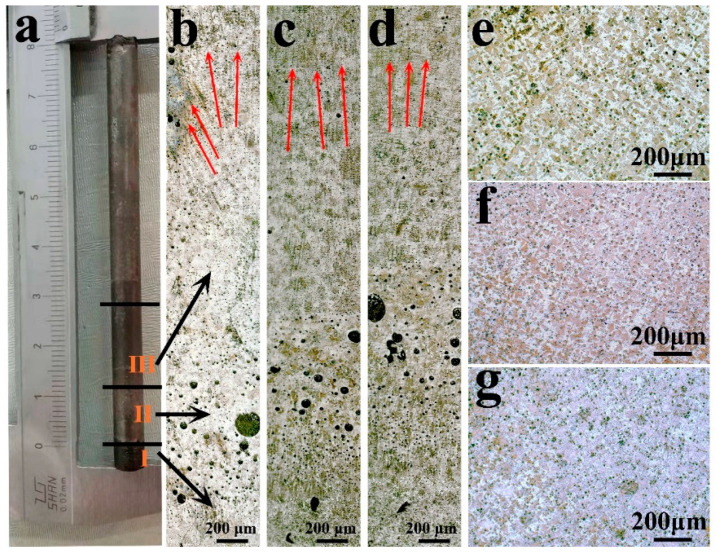
Optical microscope (OM) morphology of directional solidification (DS) CoCrCuFeNiTi_0.8_ high-entropy alloy sample. (**a**) Directional solidification samples; (**b**–**d**) are longitudinal microstructures of samples with a withdrawal rate of 50 μm/s, 100 μm/s, and 500 μm/s, respectively; (**e**–**g**) are cross-sectional microstructures of samples with a withdrawal rate of 50 μm/s, 100 μm/s, and 500 μm/s, respectively.

**Figure 2 entropy-22-00786-f002:**
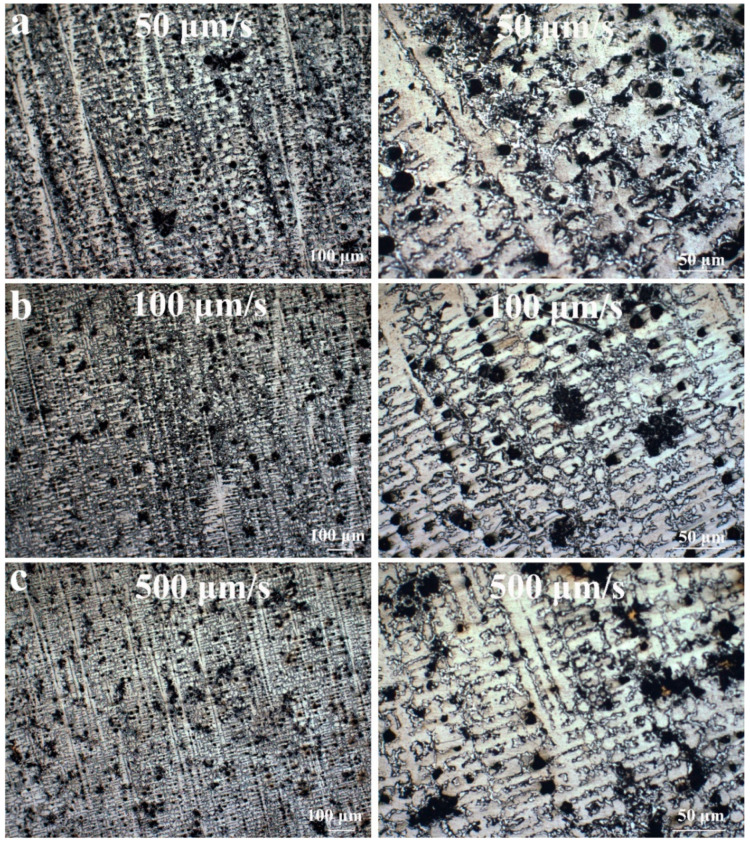
Microstructure of directional solidification stabilization zone at different extraction rates; (**a**–**c**) correspond to extraction rates of 50, 100, and 500 μm/s, respectively.

**Figure 3 entropy-22-00786-f003:**
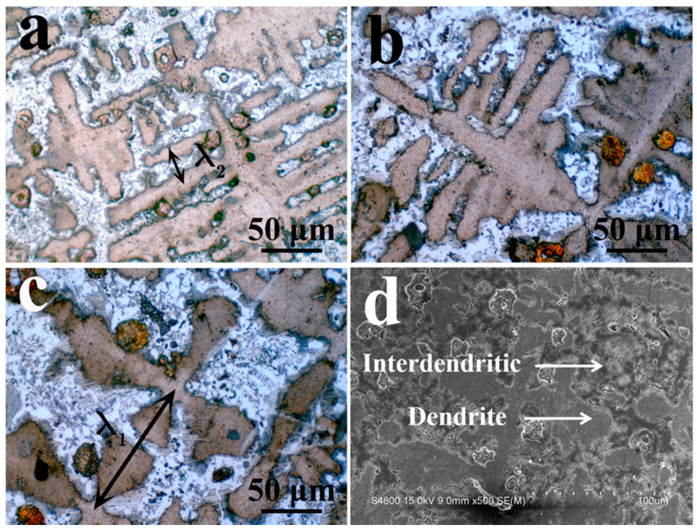
Microstructure of cross-sections of DS structures at different withdrawal rates. (**a**) Optical microscope photo at 500 μm/s. (**b**) Optical microscope photo at 100 μm/s. (**c**) Optical microscope photo at 50 μm/s. (**d**) Scanning electron microscope photo at 50 μm/s.

**Figure 4 entropy-22-00786-f004:**
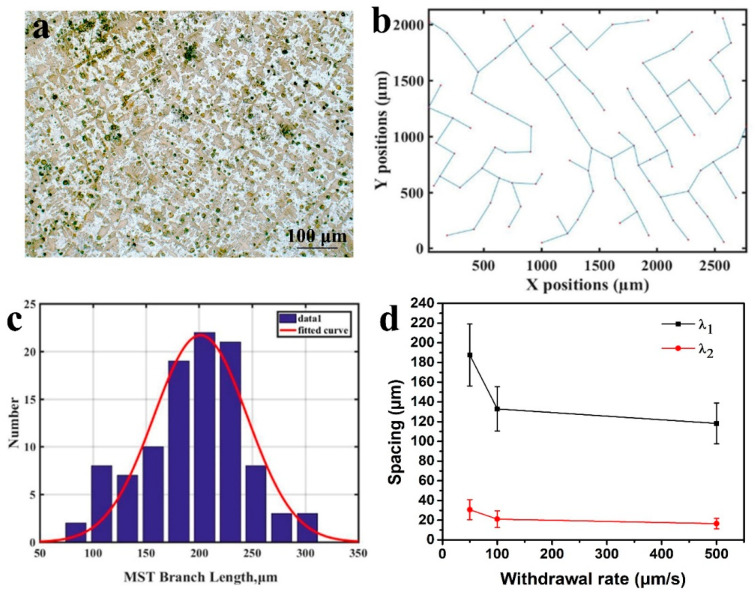
Dendrite arm spacing of DS CoCrCuFeNiTi_0.8_ high-entropy alloy. (**a**) Cross-section microstructure at 50 μm/s. (**b**) Minimum spanning tree at dendrite center. (**c**) Gaussian distribution of minimum spanning tree (MST) calculation results. (**d**) Primary and secondary dendrite spacing.

**Figure 5 entropy-22-00786-f005:**
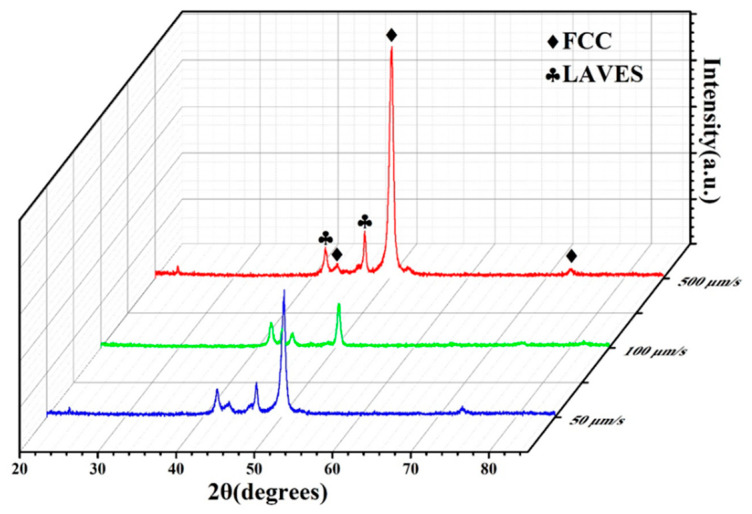
XRD spectrum of DS CoCrCuFeNiTi_0.8_ high-entropy alloy.

**Figure 6 entropy-22-00786-f006:**
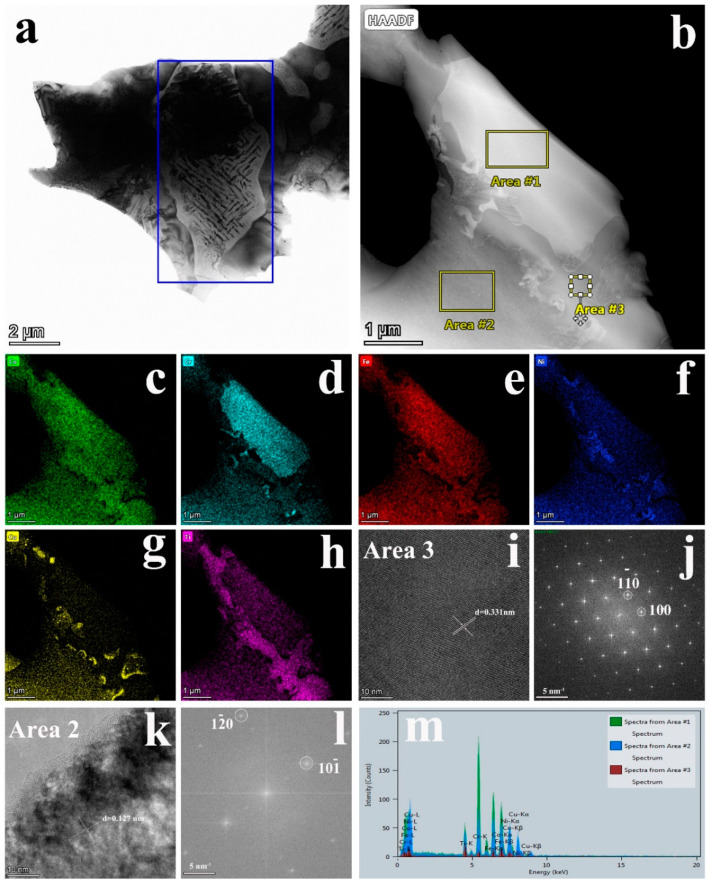
(**a**) TEM brightfield image. (**b**) TEM dark field image. (**c**–**h**) Energy dispersive spectrometry (EDS) mapping corresponding to (**b**). (**i**) High-resolution image corresponding to (**b**) Area 3. (**j**) The diffraction spots after the fast Fourier transforms (FFT) corresponding to (**i**). (**k**) High-resolution image corresponding to (**b**) Area 2. (**l**) Diffraction spots after FFT conversion corresponding to (**k**). (**m**) EDS spectrum corresponding to (**b**).

**Figure 7 entropy-22-00786-f007:**
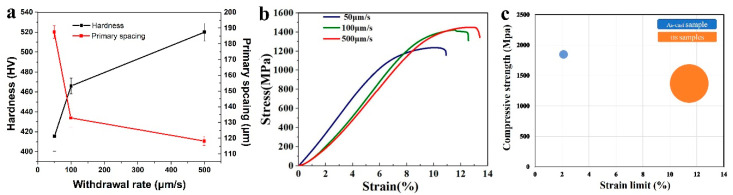
(**a**) Hardness and primary dendrite arm spacing of DS CoCrCuFeNiTi_0.8_ high-entropy alloy. (**b**) Stress–strain curves of DS CoCrCuFeNiTi_0.8_ high-entropy alloy at different withdrawal rates. (**c**) Comparison of compressive properties between DS CoCrCuFeNiTi_0.8_ alloy and non-DS CoCrCuFeNiTi_0.8_ alloy.

**Figure 8 entropy-22-00786-f008:**
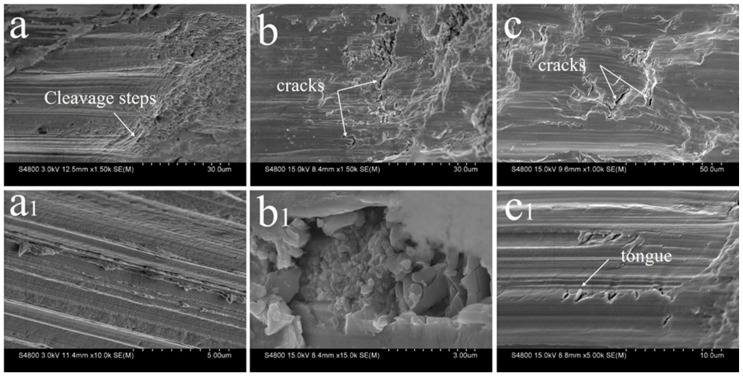
Morphologies of fracture surfaces of directionally solidified CoCrCuFeNiTi_0.8_ high-entropy alloy ((**a** and **a_1_**) 50 μm/s, (**b** and **b_1_**) 100 μm/s, (**c** and **c_1_**) 500 μm/s).

**Table 1 entropy-22-00786-t001:** Element distribution in different regions.

Regions/Element	Co	Cr	Fe	Ni	Cu	Ti
Dendrite	17.78%	20.52%	20.32%	17.12%	12.66%	11.61%
Interdendritic	21.65%	16.68%	19.18%	15.09%	7.56%	19.87%

**Table 2 entropy-22-00786-t002:** Experimental data of the room temperature compression of DS CoCrCuFeNiTi_0.8_ high-entropy alloy.

Withdrawal Rate (μm/s)	Compressive Strength/MPa	Strain Limit/%
500	1449.8	12.75
100	1421.2	11.47
50	1237.3	10.08

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
