# Peer review of "Microstructure Evolution and Mechanical Properties of FeCoCrNiCuTi0.8 High-Entropy Alloy Prepared by Directional Solidification"

_entropy, 2020, doi:10.3390/e22070786_

Round 1

Reviewer 1 Report

For their work, the authors examined the effects of withdrawal rate (during directional solidification) on the microstructure of a CoCrCuFeNiTi0.8 high entropy alloy (HEA). They report that the dendrite orientation becomes more uniform with an increase in the withdrawal rate. Although their work appears interesting, there are a few issues that need to be resolved before it can be accepted by the journal.

1. The authors should provide a more comprehensive discussion about the various aspects of high entropy alloys. Below are a few references to help with this process: 

Creep resistance: 

Y.L. Chou, Y.C. Wang, J.W. Yeh, H.C. Shih, The effect of molybdenum on the corrosion behaviour of the high-entropy alloys Co1.5CrFeNi1.5Ti0.5Mox in aqueous environments, Corrosion Science 52 (2010) 1026-1034. 

Biocompatability: 

W.-Y. Ching, S. San, J. Brechtl, R. Sakidja, M. Zhang, P.K. Liaw, Fundamental electronic structure and multiatomic bonding in 13 biocompatible high-entropy alloys, npj Comput. Mater. 6(1) (2020) 45. 

Deformation behavior: 

Y. Zhang, T.T. Zuo, Z. Tang, M.C. Gao, K.A. Dahmen, P.K. Liaw, Z.P. Lu, Microstructures and properties of high-entropy alloys, Prog. Mater. Sci. 61 (2014) 1-93.

Excellent mechanical and magnetic properties: 

C. Chen, H. Zhang, Y.Z. Fan, W.W. Zhang, R. Wei, T. Wang, T. Zhang, F.S. Li, A novel ultrafine-grained high entropy alloy with excellent combination of mechanical and soft magnetic properties, J. Magn. Magn. Mater. 502 (2020) 5. 

2. Elaborate on how the transmission electron microscopy (TEM) samples prepared. For example, how were the samples thinned?

3. For the X-ray diffraction experiment, what was the energy of the X-ray? What optics were used?

4. For the Vickers hardness tests, what ASTM standard was used? How far apart were the indents spaced?, how many indents were performed?

5. For figure 2, please put the withdrawal rate values above their corresponding figures as it is very hard to read.

6. Please use a different color other than red for the text in Fig. 3d.

7. In the conclusion section, the authors must give an overall big picture summary in a paragraph after the listed points. This will help illustrate the importance of the study to the reader.

Reviewer 2 Report

This is a good article and I think it can be published in the journal.

Maybe the article would be more beneficial if the authors added more links to studies of this particular composition (CoCrCuFeNiTi). Who investigated this composition of high entropy alloys before? What results did they get? This would be interesting to compare with the results of the authors. Even if other studies were not devoted to directed solidification of the alloy, other people's data on its structure and properties would be useful for understanding the practical value of the study.

Reviewer 3 Report

The paper is nicely done. English is good. Minor checking necessary.

Interesting paper. I have one question though from looking at Figures 1 & 2 and Figure 4: Is there porosity in these alloys or is what I see something else? To my mind I see very fine porosity distributed throughout these alloys. The shape of some of these features look like pores but I can't remember any mention in the text about this. Can the authors please clarify either in a response to this question or by modifying the narrative to include a discussion on this? If this is porosity then some indication to the amount in each draw would be very useful in interpreting the results. (I might mention that some of the higher magnification views show signs of possible abrasive pull out but Figure 1 in particular seems to show clear signs of porosity.)

I can see the possible effect of uniform fine porosity on compressive mechanical behavior both in terms of the strength of the alloy as well as the effect on plastic strain. However, what might be good in compression may not be good in tension (or cyclic behavior).

So, with respect to the above please clarify. Thank you.

One other minor point: Please don't use the term "sandpaper" as it implies something altogether different. A more appropriate term would be "abrasive" paper since then the abrasive could be anything such as SiC in this case.
